# First Report of *Kocuria kristinae* in the Skin of a Cuban Boa (*Epicrates angulifer*)

**DOI:** 10.3390/life13112133

**Published:** 2023-10-29

**Authors:** Inês Marques, Nuno Alvura, José Júlio Martins, João F. Requicha, Maria J. Saavedra

**Affiliations:** 1Department of Veterinary Sciences, University of Trás-os-Montes e Alto Douro, 5000-801 Vila Real, Portugal; inessilvamarques@gmail.com (I.M.); jfrequicha@utad.pt (J.F.R.); 2Antimicrobials, Biocides and Biofilms Unit (A2BUnit), Laboratory of Medical Microbiology, University of Trás-os-Montes e Alto Douro, 5000-801 Vila Real, Portugal; 3Zoo da Maia, Rua da Estação, 4470-184 Maia, Portugal; vet@zoodamaia.pt; 4Laboratory of Physiology, Department of Animal Science, University of Trás-os-Montes e Alto Douro, 5000-801 Vila Real, Portugal; jjulio@utad.pt; 5Animal and Veterinary Research Center (CECAV)–Al4AnimalS, University of Trás-os-Montes e Alto Douro, 5000-801 Vila Real, Portugal; 6Interdisciplinar Center of Marine and Environmental Research (CIIMAR), University of Porto, 4450-208 Matosinhos, Portugal; 7Center for the Research and Technology of Agro-Environmental and Biological Sciences (CITAB)-Inov4Agro, University of Trás-os-Montes e Alto Douro, 5000-801 Vila Real, Portugal

**Keywords:** skin, Cuban boa, *Epicrates angulifer*, *Kocuria kristinae*, One Health

## Abstract

During a routine clinical examination of a four-year-old female Cuban boa (*Epicrates angulifer*) belonging to a zoological park located in northern Portugal, a skin lesion was observed. A skin swab was taken. Bacteriological analysis conducted using the Vitek^®^ 2 Compact system identified the presence of the bacteria species *Kocuria kristinae*, a new bacterial pathogen that may be a potential pathogen in wild animals. This *K. kristinae* strain was resistant to kanamycin, pradofloxacin, erythromycin, clindamycin, tetracycline, nitrofurantoin, and trimethoprim/sulphamethoxazole and was therefore classified as a multidrug-resistant bacterium. To the authors’ knowledge, this is the first time that *K. kristinae* has been described in the skin of a Cuban boa. This report serves as a cautionary warning about the importance of recognising and investigating the potential pathogenicity of this agent, as well as contributing to the development of strategies to prevent the spread of antibiotic-resistant microorganisms.

## 1. Introduction

The Cuban boa (*Epicrates angulifer*), a member of the boid family, is native to Cuba and its neighbouring islands, and is most active at sunset and at night. As a semi-arboreal species, they seek shelter on the ground, in burrows, and under rocks and live in moist tropical forests near water [1].

They are non-venomous snakes that kill their prey via constriction and feed on mammals (especially rodents and bats) and birds. Cuban boas can reach up to 4.5 m in length and are brown with a pattern of darker spots [2].

Snakes are increasingly popular pets and major attractions in zoos, and can act as reservoirs of pathogenic bacteria for both animals and humans. Through direct or indirect contact, or even by biting, they provide excellent opportunities for the transmission of multi-resistant bacteria [3,4].

The identification of microorganisms found in snakes is important for understanding the bacterial communities that coexist with these reptiles. It also plays a crucial role in conducting antimicrobial resistance surveillance studies to improve the efficacy of the antibiotherapies used in their treatment [5,6].

The genus *Kocuria* belongs to the family Micrococcaceae, suborder Micrococcineae, order Actinomycetales, and class Actinobacteria. To date, thirty-four species have been described within the genus *Kocuria,* namely, *K. varians*, *K. rosea*, *K. kristinae*, *K. rhizophila*, *K. marina*, *K. polaris*, *K. aegyptia*, *K. carniphila*, *K. himachalensis*, *K. flava*, *K. turfanensis*, *K. atrinae*, *K. gwangalliensis*, *K. halotolerans*, *K. koreensis*, *K. coralli*, *K. salsicia*, *K. arsenatis*, *K. assamensis*, *K. dechangensis*, *K. indica*, *K. massiliensis*, *K. oceani*, *K. ocularis*, *K. palustris*, *K. pelophila*, *K. salina*, *K. sediminis*, *K. soli*, *K. subflava*, *K. tytonicola*, *K. tytonis*, *K. uropygialis*, and *K. uropygioeca* [6,7,8,9,10].

*K. kristinae* is a facultative anaerobic, Gram-positive, coccoid, non-motile, catalase-positive, and coagulase-negative bacterium. It forms non-haemolytic colonies on blood agar and is non-spore-forming, non-capsulated, non-acid-fast, and positive for Voges–Proskauer. Its optimum temperature range for growth is 25 °C to 37 °C, but it can grow at 45 °C. This bacterium frequently appears in the skin, oral cavities, and urinary tracts of mammals, including humans, and is invariably associated with immunocompromised patients, leading to severe infections [11]. It has also been described that patients with diabetes are more susceptible to *K. kristinae* infections as well as secondary infections, including retropharyngeal abscess, intracranial infection, stroke, septic pulmonary embolism, and urinary tract infection. Bacterial infections in people with diabetes result from high blood glucose levels, which can lead to compromised immune systems, such as reduced antioxidant defences, impaired neutrophil function, and decreased antibacterial defences. These changes facilitate the infiltration of pathogenic organisms and subsequently lead to the development of bacterial infections [12].

This bacterium has been isolated from various biological materials, such as blood, peritoneal fluid, pleural fluid, bile, synovial fluid, and abdominal abscesses. The most common clinical signs of *Kocuria* spp. infection are fever, sepsis, organ dysfunction, abdominal pain, and cloudy peritoneal fluid. However, to the authors’ knowledge, *K. kristinae* has not yet been reported in snakes. In domestic animals, to date, this bacterium has only been isolated from the reproductive tract of two cattle [13,14].

The aim of this study was to describe, for the first time, this microbial agent’s presence in a skin lesion of a Cuban boa.

## 2. Materials and Methods

The present study reports on a four-year-old female Cuban boa (*Epicrates angulife*) belonging to a zoological park located in northern Portugal. On physical examination, the animal presented an area with an accumulation of dry scales and extensive skin changes (Figure 1).

A skin sample was collected using a sterile AMIES swab (VWR, Carnaxide, Portugal) under manual restraint of the animal and without the need for sedation. The entire procedure was carried out in accordance with the European Animal Welfare Directives (Directive 98/58/CE and Decreto-lei 64/2000). The sampling was conducted following the established procedures for each stage of the process: collection, transport, and preservation of the sample.

Standard laboratory methods were used for bacterial isolation, purification, and identification. For microbiological analysis, specific media for the growth of Gram-negative and Gram-positive bacteria were used, in accordance with the methodologies implemented at the Antimicrobials, Biocides and Biofilms Unit (A2BUnit) at the University of Trás-os-Montes and Alto Douro and in accordance with the Laboratory Standardization of Bacterial Culture and Antimicrobial Susceptibility Testing in Veterinary Clinical Microbiology. A selective and differential chromogenic medium (Chromogenic Coliform Agar) and four selective and differential culture media (MacConkey Agar, Baird Parker Agar, Glutamate Starch Phenol Red Agar, and Mannitol Salt Agar) were used according to the manufacturer’s instructions.

After obtaining the pure cultures, isolates were selected for identification (ID) and antimicrobial susceptibility testing (AST) using the automated Vitek^®^ 2 Compact system (bioMérieux, Craponne, France). The identification of the microbial species was performed through a card consisting of a miniaturised system of conventional biochemical tests (Vitek^®^ 2 GP and Vitek^®^ 2 GN, bioMérieux, Craponne, France). The determination of antimicrobial susceptibility was performed using cards for AST composed of multiple antibiotics. In order to obtain the ID and the AST of the bacterial isolate, the protocol established by the manufacturer was followed.

Thirteen antibiotics, belonging to eight classes, were tested on Gram-positive isolates (Vitek^®^ 2 AST-GP80, bioMérieux, Craponne, France): Aminoglycosides: gentamicin (CN), kanamycin (K), and neomycin (N); Fluoroquinolones: enrofloxacin (ENR), marbofloxacin (MRB), and pradofloxacin (PFX); Macrolides: erythromycin (E); Lincosamides: clindamycin (DA); Tetracyclines: doxycycline (DO) and tetracycline (TE); Nitrofurans: nitrofurantoin (F); Phenicols: chloramphenicol (C); and Sulphamides: Sulphamethoxazole/Trimethoprim (SXT).

On the Gram-negative isolates (Vitek^®^ 2 AST-GN97, bioMérieux, Craponne, France), six antibiotics, belonging to two classes, were tested: Beta-lactams: ceftazidime (CAZ), ceftriaxone (CRO), and piperacillin (PRL); and Fluoroquinolones: danofloxacin (DAN), enrofloxacin (ENR), and marbofloxacin (MRB).

## 3. Results

The bacteriological analysis using the Vitek^®^ 2 Compact system identified one Gram-positive bacterium, *K. kristinae*, and two Gram-negative bacteria, *Stenotrophomonas maltophilia* and *Aeromonas hydrophila*.

*K. kristinae* was positive for type 1 arginine dihydrolase (ADH1), leucine araminase (LeuA), l-proline arylaminase (ProA), L-pyrrolidonyl arylamidase (PyrA), alanine araminase (AlaA), and tyrosine araminase (TyrA). Table 1 shows all the biochemical tests conducted using the Vitek^®^ 2 Compact system.

With regard to the aminoglycoside class, sensitivity to gentamicin, an intermediate level of resistance (susceptible with increased dosing) to neomycin, and resistance to kanamycin were observed. Except for the response to pradofloxacin, where the isolate showed resistance, an intermediate response to the other fluoroquinolones tested (enrofloxacin and marbofloxacin) was observed. With regard to tetracyclines, *K. kristinae* showed resistance to tetracycline and an intermediate response to doxycycline.

Resistance to erythromycin, clindamycin, nitrofurantoin, and trimethoprim-sulphamethoxazole was also observed. As for chloramphenicol, an antibiotic belonging to the phenolic class, the bacterial isolate showed an intermediate response.

It should be noted that the bacterium *K. kristinae* was resistant to seven classes of antibiotics (namely, aminoglycosides, fluoroquinolones, macrolides, lincosamides, tetracyclines, nitrofurans, and sulphamides) and was considered multi-resistant, i.e., resistant to at least one agent of three or more antimicrobial categories (Table 2).

Table 3 shows the antimicrobial susceptibility profile of *Stenotrophomonas maltophilia* and *Aeromonas hydrophila*. Both Gram-negative isolates showed resistance to beta-lactams (ceftazidime, ceftriaxone, and piperacillin) and sensitivity to fluoroquinolones (danofloxacin, enrofloxacin, and marbofloxacin).

*Stenotrophomonas maltophilia* and *Aeromonas hydrophila* can act as opportunistic pathogens and are frequently isolated from water and soil [15,16].

## 4. Discussion

The identification of multidrug-resistant bacteria in the skin of this snake is a reminder of the frequent emergence of multidrug-resistant microorganisms. This is a public health problem that can seriously undermine the ability to treat bacterial infections [17,18]. Thus, it is crucial to consider a holistic One Health approach that integrates aspects of human, animal, and environmental health in modern veterinary and medical sciences [18].

When linking these different issues, it is imperative to carry out surveillance studies on antibiotic resistance to promote greater effectiveness of the treatments implemented, as well as to assess the spread of multi-resistant bacteria in different ecological niches, including zoological parks [17,18].

In this study, we described the isolation of a new bacterial pathogen, *K. kristinae*, from the skin of a captive Cuban boa snake, which may be a potential pathogen in wild animals.

The skin acts as a physical barrier between the internal and external environment, but it also provides protection to animals against ultraviolet light from the sun [19,20]. Skin diseases are common in wild and captive animals [21].

The microorganisms present on the skin are fundamental to establishing and maintaining skin homeostasis [21]. The animal’s immune system and the skin microbiota work together to create barriers and prevent the colonisation of pathogens. The skin microbiota is therefore essential to the health of the animal, and changes in it can lead to dermatological changes. Bacterial dermatitis is a common problem in reptiles [22].

It is well established that *K. kristinae* is a common organism on the skin, oral cavity, and urinary tract of mammals, and it has been isolated from cornea, pleural effusion, and peripheral blood samples in humans [13]. It has been implicated as an aetiological agent in several human diseases, including endocarditis, peritonitis, and pneumonia. It can also cause septicaemia in immunosuppressed people [11].

The antibiotic susceptibility profile of *K. kristinae* shows that we are dealing with a multidrug-resistant bacterium, as it shows resistance to aminoglycosides, fluoroquinolones, macrolides, lincosamides tetracyclines, nitrofurans, and sulphamides.

In the present work, thirteen antibiotics belonging to eight classes were tested; *K. kristinae* showed resistance to seven of them and sensitivity only to gentamicin.

*Aeromonas* spp. are Gram-negative bacilli commonly isolated from soil and water. The most health-relevant bacterium is *Aeromonas hydrophila*, which is associated with ulcerative inflammation in the skin of frogs, toads, and fish [23,24]. *Aeromonas hydrophila* is regularly found in raw surface groundwater and has been detected in drinking water supplies [15]. It is an opportunistic bacterial pathogen and can cause severe disease and septicaemia [25,26].

Members of the genus *Aeromonas* are ubiquitous in aquatic environments and they cause a wide range of opportunistic infections in freshwater and brackish water fish. *Aeromonas* are also emerging human pathogens and have been shown to form biofilms on a variety of biotic and abiotic surfaces [24,26,27].

*Stenotrophomonas maltophilia* is an opportunistic Gram-negative pathogenic bacterium with high antimicrobial resistance [16,28]. This bacterium is found in various environmental sources, including plants, soil, animals, and aquatic environments. *Stenotrophomonas maltophilia* is associated with ulcerative stomatitis in snakes and septicaemia in crocodiles, and is the only species of the *Stenotrophomonas* genus that can cause infections in humans, mainly in hospitalised patients and immunocompromised patients [29,30,31].

Most of the infections caused by *Stenotrophomonas maltophilia* occur in the lower respiratory tract, but it can also cause other infections, such as meningitis, urinary tract infections, peritonitis, and wound and soft tissue infections [30].

Both Gram-negative isolates showed resistance to beta-lactams and sensitivity to fluoroquinolones. The microorganisms present on the skin of these animals are usually found in their environment, such as in the water and soil. On the other hand, the intestinal tract of the snakes themselves can also be colonised by a variety of bacteria which can contaminate their water sources and the soil in their enclosures.

Daily care and feeding of these animals by their keepers may be sufficient for the transmission and spread of pathogens. Therefore, it is of paramount importance to prevent the transmission of pathogenic microorganisms by using specific equipment for cleaning enclosures and thorough hand washing after contact with animals [18,32].

It is important to note that we should not rule out the possibility that the newly detected agent was transmitted via animal keepers or food, or through contact with terrarium surfaces. To mitigate the hazardous potential of pathogenic antibiotic-resistant bacteria transferring from animals to the environment, it is imperative to implement laboratory diagnostics.

*K. kristinae* may not have been the cause of the clinical lesion but it may have acted as an opportunistic bacterium. It is therefore important to further characterise the bacterial aetiology of skin lesions in snakes. Some of these pathogens have the ability to present resistance genes to the antibiotics used in the prescribed treatment. Infectious diseases, often caused by multidrug-resistant bacteria, are a major cause of morbidity and mortality. In the future, it will be important to identify the genetic determinants of resistance in order to better understand the mechanisms of resistance. Antimicrobial resistance is one of the most important public health challenges, in which the clinical microbiology laboratory plays a critical role by providing guidance for antimicrobial treatment.

## 5. Conclusions

The excessive and inappropriate use of antibiotics in human and veterinary medicine has led to the spread of antibiotic-resistant microorganisms. This escalating level of antibiotic resistance poses a significant threat to both animal and public health. Therefore, laboratory diagnostics play a critical role in identifying emerging pathogens and conducting antibiotic resistance surveillance studies. By using robust diagnostic techniques, microbiologists can accurately identify new pathogens affecting animals. This identification is essential for the implementation of appropriate treatment strategies and preventive measures.

The potential transmission of antibiotic-resistant microorganisms from animals to humans highlights the link between veterinary and human medicine. Timely and accurate clinical diagnosis in veterinary microbiology is essential to understanding and managing the risks associated with antimicrobial resistance in wildlife.

This report serves as a cautionary alert to the importance of recognising and investigating the potential pathogenicity of *K. kristinae*, as well as its multi-resistance profile. To the authors’ knowledge, this is the first description of this potential bacterial pathogen present in the skin of a Cuban boa (*Epicrates angulifer*).

It is crucial not to ignore or underestimate the presence of this bacterium when it is isolated and identified in different biological samples. Vigilance in recognising and thoroughly investigating the presence of this bacterium in biological samples is essential for effective management and treatment strategies.

## Figures and Tables

**Figure 1 life-13-02133-f001:**
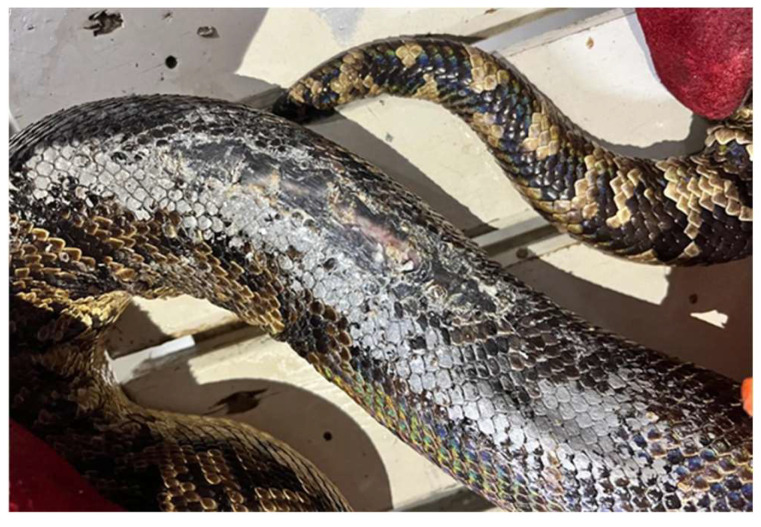
Macroscopic aspect of the skin wound observed in the studied Cuban boa.

**Table 1 life-13-02133-t001:** Results of the characterisation of *K. kristinae* by biochemical tests (Vitek^®^ 2 GP).

*AMY*	−	*PIPLC*	−	*dXYL*	−	*ADH1*	+	*BGAL*	−	*AGLU*	−
*APPA*	−	*CDEX*	−	*AspA*	−	*BGAR*	−	*AMAN*	−	*PHOS*	−
*LeuA*	+	*ProA*	+	*BGURr*	−	*AGAL*	−	*PyrA*	+	*BGUR*	−
*AlaA*	+	*TyrA*	+	*dSOR*	−	*URE*	−	*POLYB*	−	*dGAL*	−
*dRIB*	−	*ILATk*	+	*LAC*	−	*NAG*	−	*dMAL*	−	*BACI*	−
*NOVO*	−	*NC6.5*	−	*dMAN*	−	*dMNE*	−	*MBdG*	−	*PUL*	−
*dRAF*	−	*O129R*	−	*SAL*	−	*SAC*	−	*dTRE*	−	*ADH2s*	−
*OPTO*	−										

Legend: AMY: d-amygdalin; PIPLC: phosphatidylinositol phospholipase C; dXYL: d-xylose; ADH1: arginine dihydrolase 1; BGAL: beta-galactosidase; AGLU: alpha-glucosidase; APPA: Ala-Fe-Pro arylamidase; CDEX: cyclodextrin; AspA: l-aspartate arylamidase; BGAR: beta galactopyranosidase; AMAN: alpha-mannosidase; PHOS: phosphatase; LeuA: leucine arylamidase; ProA: l-proline aryldaminase; BGURr: beta-glucuronidase; AGAL: alpha-galactosidase; PyrA: l-pyrrolidonyl arylamidase; BGUR: beta-glucuronidase; AlaA: alanine arylamidase; TyrA: tyrosine arylamidase; dSOR: d-sorbitol; URE: urease; POLYB: polymyxin B resistance; dGAL: d-galactose; dRIB: d-ribose; ILATk: l-lactate alkalinisation; LAC: lactose; NAG: N-acetyl-d-glucosamina; dMAL: d-maltose; BACI: bacitracin resistance; NOVO: resistance to novobiocin; NC6.5: growth in sodium chloride (NaCl) 6.5%; dMAN: d-mannitol; dMNE: d-mannose; MBdG: methyl-B-d-glucuryanoside; PUL: pullulan; dRAF: d-raffinose; O129R: O/129 resistance; SAL: salicin; SAC: sucrose; dTRE: d-trehalose; ADH2s: arginine dihydrolase 2; OPTO: optochin resistance; +: positive; −: negative.

**Table 2 life-13-02133-t002:** Antibiotic susceptibility profile of *K. kristinae* (Vitek^®^ 2 AST-GP80).

Antibiotic	MIC	Interpretation
Gentamicin (CN)	≤0.5	Susceptible
Kanamycin (K)	32	Resistance
Neomycin (N)	16	Intermediate
Enrofloxacin ENR)	2	Intermediate
Marbofloxacin (MRB)	2	Intermediate
Pradofloxacin (PFX)	≥4	Resistance
Erythromycin (E)	≥8	Resistance
Clindamycin (DA)	≥4	Resistance
Doxycycline (DO)	8	Intermediate
Tetracycline (TE)	≥16	Resistance
Nitrofurantoin (F)	128	Resistance
Chloramphenicol (C)	16	Intermediate
Trimethoprim/sulphamethoxazole (SXT)	≥320	Resistance

Legend: Aminoglycosides: CN, K, N; Fluoroquinolones: ENR, MRB, PFX; Macrolides: E; Lincosamides: DA; Tetracyclines: DO, TE; Nitrofurans: F; Phenicols: C; Sulphamides: SXT.

**Table 3 life-13-02133-t003:** Antibiotic susceptibility profile of *Stenotrophomonas maltophilia* and *Aeromonas hydrophila* (Vitek^®^ 2 AST-GN97).

Antibiotic/Isolate	*Stenotrophomonas maltophilia*	*Aeromonas hydrophila*
Ceftazidime (CAZ)	Resistance	Resistance
Ceftiaxone (CRO)	Resistance	Resistance
Piperacillin (PRL)	Resistance	Resistance
Danofloxacin (DAN)	Susceptible	Susceptible
Enrofloxacin (ENR)	Susceptible	Susceptible
Marbofloxacin (MRB)	Susceptible	Susceptible

Legend: Beta-lactams: CAZ, CRO, PRL; Fluoroquinolones: DAN, ENR, MRB.

## Data Availability

Not applicable.

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
