# Peer review of "First Report of Kocuria kristinae in the Skin of a Cuban Boa (Epicrates angulifer)"

_life, 2023, doi:10.3390/life13112133_

Round 1
Reviewer 1 Report
Comments and Suggestions for Authors
The study reported skin lesion on a four-year-old Cuban boa at a zoological park in northern Portugal. The analysis revealed a new bacterial pathogen,
Kocuria kristinae, which can be harmful to wild animals; also, the bacterium is multidrug-resistant and was found for the first time in a Cuban boa. The report warns about recognizing and investigating potential pathogens to prevent the spread of antibiotic-resistant microorganisms. The study explores an interesting subject; however, there are some notable
shortcomings in the manuscript that needs authors attention and correction.
1. The author reported the collection of dry skin samples and raised questions about the bacteria's ability to survive in such conditions. I intrigued by the mechanisms enabling this survival. Is the bacteria heat-resistant? Could you elaborate on the specific physiological conditions necessary for the bacteria to thrive in this environment? The inclusion of details regarding the bacteria's physiological requirements would be valuable.
2. The author noted a pronounced vulnerability to K. kristinae infection among individuals afflicted with diabetes. Author suggested to elucidate comprehensively the intricate association between diabetes and bacterial infections.
3. Author mentioned that K. kristinae is a multidrug-resistant strain. Does this type of bacterial infection have a serious impact on wild animals? Are there any previous studies related to this type of infection?
4. Author confirms the drug resistance of K. kristinae in biochemical tests. The author suggests focusing on molecular confirmation for more accurate results compared to biochemical tests.
5. Is there any literature available on Kocuria species that cause clinical symptoms? The author suggests explaining the various clinical symptoms since it is multidrug-resistant bacteria.
6. The method described by the author for isolating K. kristinae from the skin raises concerns due to the typical presence of diverse microbial flora on the skin. It remains unclear how the author successfully isolated this specific bacterium amidst the variety of microorganisms usually found on the skin. Although the author mentioned the utilization of specific media for distinguishing between gram-positive and gram- negative bacteria, additional details on the screening techniques employed to isolate
this particular strain would enhance the clarity and reliability of the methodology.
Author Response
Dear Reviewer 1,
In response to the review of the paper entitled “First report of Kocuria kristinae in the skin from Cuban Boa (Epicrates angulifer)", we would like to thank you for your insightful suggestions. In this version, we have attended to all the comments and suggestions to improve the quality of the manuscript. All the required changes have been improved and supplemented with more relevant information, considering the comments of the reviewers.
We found the comments and suggestions very useful to improve the quality of the study and make it clearer, and thus we hope all the modifications made to the manuscript are now clear and concise enough as required to enable the publication of the manuscript.
Sincerely Yours,
Inês Marques, Nuno Alvura, José Júlio Martins, João F. Requicha, Maria J. Saavedra
Response to Reviewer 1/ Comments
The study reported skin lesion on a four-year-old Cuban boa at a zoological park in northern Portugal. The analysis revealed a new bacterial pathogen,
Kocuria kristinae, which can be harmful to wild animals; also, the bacterium is multidrug-resistant and was found for the first time in a Cuban boa. The report warns about recognizing and investigating potential pathogens to prevent the spread of antibiotic-resistant microorganisms. The study explores an interesting subject; however, there are some notable shortcomings in the manuscript that needs authors attention and correction.
1. The author reported the collection of dry skin samples and raised questions about the bacteria's ability to survive in such conditions. I intrigued by the mechanisms enabling this survival. Is the bacteria heat-resistant? Could you elaborate on the specific physiological conditions necessary for the bacteria to thrive in this environment? The inclusion of details regarding the bacteria's physiological requirements would be valuable.
Response 1: We thank the reviewer for this suggestion. We fully agree and the information has been added with more detail.
Kocuria spp. is widely distributed in nature, its optimum temperature range for growth is 25˚C to 37˚C, but it can grow at 45˚C. Other biochemical and physiological properties of Kocuria spp. are the formations of non-hemolytic colonies on blood agar, non-spore-forming, non-capsulated, non-acid-fast and positive for Voges-Proskauer test.
- The author noted a pronounced vulnerability to K. kristinae infection among individuals afflicted with diabetes. Author suggested to elucidate comprehensively the intricate association between diabetes and bacterial infections.
Response 2: Thank you for your comment. This information would be interesting and helpful to the readers.
Diabetes predisposes to a number of complications in the body, including a weakened immune system that impairs the resolution of infectious conditions. Bacterial infections in individuals with diabetes result from high blood glucose levels that can lead to compromised immune systems, such as reduced antioxidant defenses, impaired neutrophil function, and decrease in antibacterial defenses. These changes facilitate the infiltration of pathogenic organisms and subsequently lead to the emergence of bacterial infections.
- Author mentioned that K. kristinae is a multidrug-resistant strain. Does this type of bacterial infection have a serious impact on wild animals? Are there any previous studies related to this type of infection?
Response 3: Thank you for your comment.
To the best of our knowledge, K. kristinae has not yet been reported in wildlife. Wild animals are of great importance in the One Health concept. They represent a potential natural reservoir of AMR and infection. They are also actively involved in the spread of bacteria and resistance determinants between different habitats.
The bacterial agent may have been transmitted by animal keepers, food or through contact with terrarium surfaces. Emerging infectious diseases often result from the interplay between wildlife, humans, and the environment. These findings highlight the importance of personal protective equipment for a variety of roles, including animal keepers, veterinarians and healthcare professionals.
This Gram-positive bacterium, primarily considered as a commensal microorganism in human skin, has recently been associated with opportunistic infections in immunocompromised humans. In domestic animals, to date, this bacterium has so far only been isolated from the reproductive tract.
- Author confirms the drug resistance of K. kristinae in biochemical tests. The author suggests focusing on molecular confirmation for more accurate results compared to biochemical tests.
Response 4: Thank you for your comment. Rapid and accurate detection of multidrug resistant bacteria is important not only for the early initiation of appropriate antibiotic therapy, but also for minimising subsequent antimicrobial resistance.
Infectious diseases, often caused by multidrug-resistant bacteria, are a major cause of morbidity and mortality. Antimicrobial resistance is one of the most important public health challenges in which the clinical microbiology laboratory plays a critical role by providing guidance on antimicrobial therapy.
- Is there any literature available on Kocuria species that cause clinical symptoms? The author suggests explaining the various clinical symptoms since it is multidrug-resistant bacteria.
Response 5: Thank you for your recommendation. We agree and we improve the document.
The most common clinical presentation of skin and soft tissue infection by Kocuria spp are fever, sepsis and organ dysfunction. Cloudy peritoneal fluid at and abdominal pain and fever are de the most common signs of peritonitis associated to Kocuria spp infection.
- The method described by the author for isolating K. kristinae from the skin raises concerns due to the typical presence of diverse microbial flora on the skin. It remains unclear how the author successfully isolated this specific bacterium amidst the variety of microorganisms usually found on the skin. Although the author mentioned the utilization of specific media for distinguishing between gram-positive and gram- negative bacteria, additional details on the screening techniques employed to isolate this particular strain would enhance the clarity and reliability of the methodology.
Response 6: Thank you for your comment. We completely agree and changed accordingly to a more understandable form.
For microbiological analysis were used specific media for the growth of Gram-negative and Gram-positive bacteria according to the methodologies implemented at the Antimicrobials, Biocides and Biofilms Unit (A2BUnit), Medical Microbiology Laboratory at the University of Trás-os-Montes and Alto Douro, according laboratory standardization of bacterial culture and antimicrobial susceptibility testing in Veterinary Clinical Microbiology. Selective, differential and chromogenic media was used, according to the manufacturer’s instructions. The shape and colour of the colonies were observed. After obtaining the pure cultures, the identification of the bacterial isolates and the antimicrobial sensitivity test (ID/TSA test) were carried out using the automated Vitek® 2 Compact system (bioMérieux, Marcy-l'Étoile, France).

Reviewer 2 Report
Comments and Suggestions for Authors
The article titled “First report of Kocuria kristinae in the skin from Cuban Boa 2 (Epicrates angulifer)” in my opinion is a valuable paper. Emerging infectious diseases often result from the interplay between wild animals, humans, and the environment. The discovery of previously unrecognized pathogens and the resurgence of known ones can lead to challenging diagnostic and management issues. In the realm of veterinary medicine, especially concerning wild animals, understanding and identifying new pathogens is pivotal. This importance stems from the potential of these pathogens to cross species barriers and also their ability to develop resistance to standard treatments.
A case in point is the observation made during a routine clinical examination of a Cuban boa (Epicrates angulifer) in a zoological park situated in northern Portugal. The discovery of a skin lesion on the snake led to further diagnostic endeavours, unearthing the presence of the bacteria species Kocuria kristinae. This Gram-positive bacterium, primarily considered as a commensal microorganism in human skin, has lately been associated with opportunistic infections in immunocompromised humans. The revelation of its presence in a Cuban boa, coupled with its multidrug-resistant nature, underscores the dynamic and often unpredictable world of microbial pathogens.
The documentation of Kocuria kristinae in the skin of a Cuban boa not only extends our understanding of the bacterium's host range but also calls attention to the larger concerns of antibiotic resistance among wildlife pathogens. Such findings serve as a stark reminder of the intricate and interwoven connections between human, animal, and environmental health. This article seeks to detail the discovery, identification, and implications of Kocuria kristinae in a Cuban boa, highlighting the urgency of holistic, One Health approaches in modern veterinary and medical sciences.
I have only minor comments:
1. Usage of Bacterial Name: It's paramount for clarity that when introducing scientific names in academic and research literature, especially when referring to them frequently, a standard procedure is followed. For the bacterium "Kocuria kristinae," the authors are recommended to:
- On its first mention in the text, introduce the bacterium with its full name followed by its abbreviation in parentheses. For example, "Kocuria kristinae (K. kristinae)."
- Subsequent mentions in the paper should use only the abbreviation "K. kristinae."
This consistent approach will aid in ensuring clarity and conciseness for readers, minimizing any potential confusion.
Line 54: Kocuria
2. Verification of Species Data:
It's crucial in scientific literature to present accurate data, especially when stating specific details like the number of species within a genus. I observed a discrepancy regarding the number of species under the genus "Kocuria."
- Authors are advised to recheck primary literature sources, consult current microbiological databases, and possibly collaborate with experts in the field to ensure accuracy.
- https://www.ncbi.nlm.nih.gov/Taxonomy/Browser/wwwtax.cgi?id=57493
- If multiple sources present varying data, it's beneficial to mention the most widely accepted or recent data, while noting the discrepancy for clarity.
3. Table 1:
If a table in your document lacks headers, its contents might not be immediately clear to readers. Headers are essential for quickly conveying what kind of information the columns and rows of a table contain. However, if you feel a table format isn't best for your information, here are a few alternative ways to present data: infographics, flowcharts or charts and graphs.
Whatever method The Authors choose, ensure that the data's integrity remains intact and that the information is easily understandable by the readers. If possible, gather feedback on your chosen format to confirm clarity and comprehension.
By addressing the above points, the manuscript can achieve a higher level of accuracy and clarity, ensuring it meets the rigorous standards of scientific publishing.
Author Response
Dear Reviewer 2,
In response to the review of the paper entitled “First report of Kocuria kristinae in the skin from Cuban Boa (Epicrates angulifer)", we would like to thank you for your insightful suggestions. In this version, we have attended to all the comments and suggestions to improve the quality of the manuscript. All the required changes have been improved and supplemented with more relevant information, considering the comments of the reviewers.
We found the comments and suggestions very useful to improve the quality of the study and make it clearer, and thus we hope the modifications made to the manuscript are now clear and concise enough as required to enable the publication of the manuscript.
Sincerely Yours,
Inês Marques, Nuno Alvura, José Júlio Martins, João F. Requicha, Maria J. Saavedra
Response to Reviewer 2/ Comments
The article titled “First report of Kocuria kristinae in the skin from Cuban Boa (Epicrates angulifer)” in my opinion is a valuable paper. Emerging infectious diseases often result from the interplay between wild animals, humans, and the environment. The discovery of previously unrecognized pathogens and the resurgence of known ones can lead to challenging diagnostic and management issues. In the realm of veterinary medicine, especially concerning wild animals, understanding and identifying new pathogens is pivotal. This importance stems from the potential of these pathogens to cross species barriers and also their ability to develop resistance to standard treatments.
A case in point is the observation made during a routine clinical examination of a Cuban boa (Epicrates angulifer) in a zoological park situated in northern Portugal. The discovery of a skin lesion on the snake led to further diagnostic endeavours, unearthing the presence of the bacteria species Kocuria kristinae. This Gram-positive bacterium, primarily considered as a commensal microorganism in human skin, has lately been associated with opportunistic infections in immunocompromised humans. The revelation of its presence in a Cuban boa, coupled with its multidrug-resistant nature, underscores the dynamic and often unpredictable world of microbial pathogens.
The documentation of Kocuria kristinae in the skin of a Cuban boa not only extends our understanding of the bacterium's host range but also calls attention to the larger concerns of antibiotic resistance among wildlife pathogens. Such findings serve as a stark reminder of the intricate and interwoven connections between human, animal, and environmental health. This article seeks to detail the discovery, identification, and implications of Kocuria kristinae in a Cuban boa, highlighting the urgency of holistic, One Health approaches in modern veterinary and medical sciences.
I have only minor comments:
- Usage of Bacterial Name:It's paramount for clarity that when introducing scientific names in academic and research literature, especially when referring to them frequently, a standard procedure is followed. For the bacterium "Kocuria kristinae," the authors are recommended to:
- On its first mention in the text, introduce the bacterium with its full name followed by its abbreviation in parentheses. For example, "Kocuria kristinae (K. kristinae)."
- Subsequent mentions in the paper should use only the abbreviation "K. kristinae."
This consistent approach will aid in ensuring clarity and conciseness for readers, minimizing any potential confusion.
Line 54: Kocuria
Response 1: We thank the reviewer for this suggestion. We completely agree and changes were made accordingly.
- Verification of Species Data:
It's crucial in scientific literature to present accurate data, especially when stating specific details like the number of species within a genus. I observed a discrepancy regarding the number of species under the genus "Kocuria."
- Authors are advised to recheck primary literature sources, consult current microbiological databases, and possibly collaborate with experts in the field to ensure accuracy.
- https://www.ncbi.nlm.nih.gov/Taxonomy/Browser/wwwtax.cgi?id=57493
- If multiple sources present varying data, it's beneficial to mention the most widely accepted or recent data, while noting the discrepancy for clarity.
Response 2: Thank you for the observation. In fact, thirty-four species have been described within the genus Kocuria. The number of species is now corrected.
- Table 1:
If a table in your document lacks headers, its contents might not be immediately clear to readers. Headers are essential for quickly conveying what kind of information the columns and rows of a table contain. However, if you feel a table format isn't best for your information, here are a few alternative ways to present data: infographics, flowcharts or charts and graphs.
Whatever method The Authors choose, ensure that the data's integrity remains intact and that the information is easily understandable by the readers. If possible, gather feedback on your chosen format to confirm clarity and comprehension.
By addressing the above points, the manuscript can achieve a higher level of accuracy and clarity, ensuring it meets the rigorous standards of scientific publishing.
Response 3: We thank the reviewer for this suggestion. Headers (Table 1, Table 2 and Table 3) were improved to better understanding.
